# Position: Current Model Cards Are Insufficient for Downstream Governance of Open-Weight Foundation Models

Sungwon Chae [1]   Keonwoo Kim [1 2]   Hoki Kim [3]   Jaeyeon Ju [1]   Sangchul Park [1]

## Abstract

The growth of open-weight foundation models (OWFMs) has prompted the AI community to re-evaluate strategies for effective downstream governance. Although model cards have been widely adopted as transparency artifacts in model repositories, existing frameworks often fail to adequately inform downstream developers and users about the distinct safety challenges posed by OWFMs. This position paper analyzes 500 model cards hosted on Hugging Face and argues that effective governance of OWFMs requires a multi-layered approach integrating three complementary components: (i) model cards, (ii) acceptable use policies (AUPs), and (iii) licenses. To motivate this claim, we identify a safety gap left by existing regulatory approaches, including model heritage, alignment provenance, and empirically observed behaviors, through an analysis of model cards with safety-critical information. We further argue that standard open-source licenses (OSLs) are not well suited for OWFMs and may weaken the enforceability of AUPs. Building on these observations, we outline directions for evolving model cards, AUPs, and licenses into integrated safety artifacts to enable a more comprehensive governance framework that coherently integrates informational, normative, and legal dimensions.

## 1. Introduction

Most open foundation models fall into the category of *open-weight* models rather than *open-source* models, in that only model weights, rather than training codes and data, are made publicly available. Open-weight foundation models (OWFMs) are commonly released and distributed through model repositories such as Hugging Face, which enable extensive downstream development and deployment. While openness can improve transparency, reproducibility, and innovation, it also expands the surface for downstream misuse beyond the control of upstream developers, including public security threats such as abuse by non-state actors (National Telecommunications and Information Administration, 2024).

One of the most significant advances in addressing these challenges was the development of the model card framework by Mitchell et al. (2019) while at Google. Following their move to Hugging Face, the platform implemented model cards as a Markdown file with a YAML header containing model metadata (Hugging Face, 2025). In practice, model card implementations have largely emphasized transparency around performance characteristics, while sections addressing bias, risks, and limitations, even if included in standard templates, remain optional and are often underdeveloped (Mitchell et al., 2019; Oreamuno et al., 2024; Liang et al., 2024; Jiang et al., 2023b). Moreover, while regulatory frameworks such as the EU AI Act (European Union, 2024) and California's Transparency in Frontier Artificial Intelligence Act (State of California, 2025) mandate the disclosure of safety risks, they primarily apply to "frontier AI" models that exceed specified training compute thresholds ($10^{25}$ FLOPs and $10^{26}$ integer OPs/FLOPs, respectively).

However, the current versions of model cards are not enough. While they can facilitate information flows, they cannot establish normative boundaries or allocate responsibility between upstream and downstream actors. In response, some developers have introduced Acceptable Use Policies (AUPs) to define prohibited or restricted downstream uses. However, the literature questions their practical effectiveness and legitimacy, noting (i) fragmented standards, (ii) unilateral norm-setting by private actors, and (iii) weak enforceability absent clear legal incorporation (Klyman, 2024). Moreover, AUPs are often missing from or insufficiently specified in model documentation and are generally not legally enforceable unless incorporated into binding licensing terms.

This limitation of AUPs is amplified by their potential conflict with model licensing. While most developers disclose an applicable license in model cards, widely used open-

---

[1]Seoul National University [2]NAVER Applied AI Group [3]Chung-Ang University. Correspondence to: Sangchul Park <parks@snu.ac.kr>.

*Proceedings of the 43rd International Conference on Machine Learning*, Seoul, South Korea. PMLR 306, 2026. Copyright 2026 by the authors.

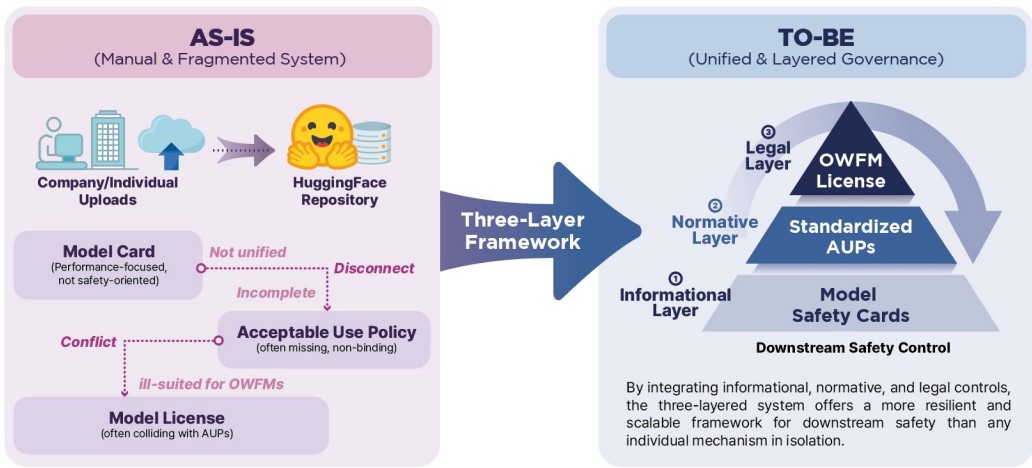

*Figure 1.* **A Three-Layered Approach to Downstream Governance of Open-Weight Foundation Models. (Left) AS-IS**: The current fragmented system, where model cards focus on performance rather than safety, AUPs are often missing or non-binding, and model licenses frequently conflict with stated usage restrictions. **(Right) TO-BE**: The proposed unified framework integrating three complementary layers for more resilient downstream safety control: informational (safety cards), normative (standardized AUPs), and legal (OWFM-tailored licenses).

source licenses (OSLs), such as Apache and MIT licenses, are structurally ill-suited to OWFMs: they neither accommodate use-based restrictions nor reflect the technical realities of model reuse, fine-tuning, and redistribution (Duan et al., 2025; McDuff et al., 2024; Contractor et al., 2022). By granting broad, unconditional rights to use, modify, and redistribute the model, permissive OSLs leave little doctrinal room for downstream use restrictions.

Building on this background, this paper asks: *How have OWFM developers operationalized downstream governance? What limitations do existing mechanisms exhibit? And how should the current framework be reformed?*

In response, this position paper proposes **strengthening downstream governance for OWFMs along three layers: model cards (informational governance), AUPs (normative governance), and model licensing (legal governance).** Each layer operates through a different mechanism of downstream control. Treating these concepts as interchangeable, or simply grouping them under "transparency," hides their important differences and may weaken effective governance. Accordingly, this paper calls for **(i) incorporating explicit safety-related components into model cards**; **(ii) developing standardized AUP templates and practices**; and **(iii) evolving existing OSLs into OWFM-tailored licensing schemes**, as illustrated in Figure 1.

This paper makes the following contributions:

1. Surveying existing model documentation practices and identifying their safety gaps and limitations.
2. Introducing a safety card template for open repositories that incorporates heritage disclosure, alignment provenance, and operational safety evaluation.

3. Proposing an alternative model licensing scheme that integrates OWFM-tailored licensing with AUPs, as an alternative to OSLs.

## 2. Model Documentation Practices

### 2.1. Overview

In open model repositories, model cards have emerged as the primary artifact for transmitting model properties to downstream. Their primary role is to describe a model's capabilities, limitations, and risks, supporting transparency and standardized research communication. To characterize prevailing documentation practices in OWFM ecosystems, we analyze the 500 most-downloaded models and their associated model cards on Hugging Face (as of January 11, 2026). Our analysis specifically examines three core governance components: (i) the occurrence of safety-related keywords in model cards, based on the taxonomies defined in NIST (2024) (see Appendix II for the full list), (ii) AUPs, and (iii) licensing structures. Although a Usage Policy for Qwen has been announced by Alibaba (2025), we excluded it from our AUP count because it is not referenced in the model cards or licensing terms. Table 1 summarizes the prevalence of these artifacts across the dataset.

*Table 1.* **Governance artifact presence in top 500 OWFMs.**

| Artifact Type | Count | Percentage |
|---|---|---|
| Model card (any form) | 498 | 99.6% |
| Safety-specific fields | 376 | 75.2% |
| AUPs | 106 | 21.2% |
| Explicit license | 425 | 85.0% |

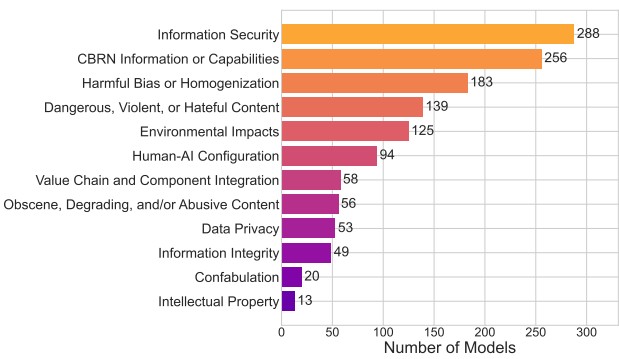

Figure 2. **Prevalence of Safety-Related Keywords Across the Top 500 Downloaded Models.** Each bar represents the number of models whose documentation contains at least one keyword from the corresponding safety category, based on the NIST GenAI Risk Management Framework taxonomy.

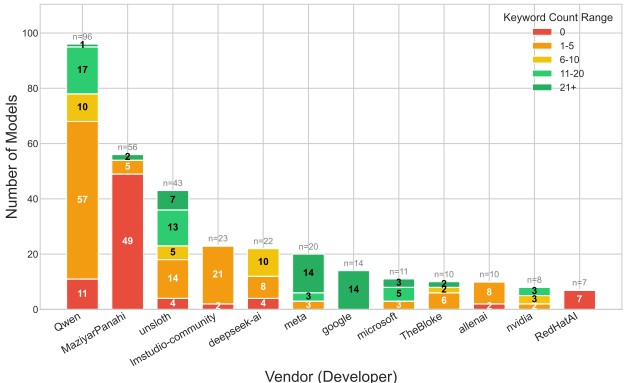

Figure 3. **Safety-Related Keyword Coverage by Vendor (Top 12 by Model Count).** Stacked bars show the number of models in each keyword-count range (0, 1–5, 6–10, 11–20, 21+). Vendors such as meta (including Llama and facebook OPT models) and google exhibit higher keyword density, while Qwen models are concentrated in lower-frequency ranges, highlighting systematic differences in disclosure practices.

Our findings reveal several deficiencies in how OWFMs are currently governed:

- *Pervasive Incompleteness*: While 99.6% of models provide a model card, the quality and structure of documentation are highly uneven. Only 75.2% include safety-specific fields, and safety information is frequently buried within generic README files rather than presented as structured, standalone disclosures.
- *AUPs*: Although explicit licensing is common, applying to 85.0% of models, AUPs remain largely absent, present in only 21.2%. Most models rely on standard, and often permissive OSLs without accompanying normative usage constraints.
- *Licensing Conflicts*: Licensing terms frequently omit safety considerations entirely.

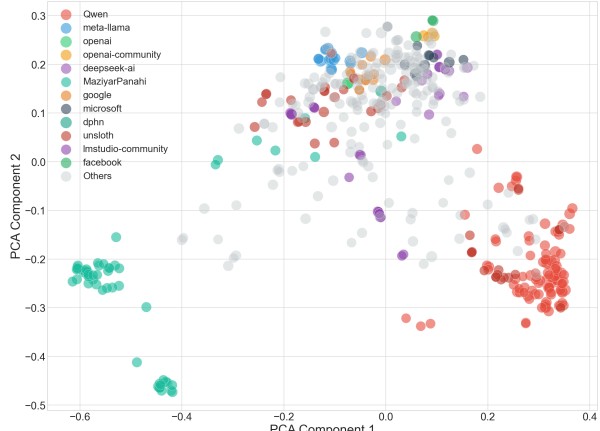

Figure 4. **Vendor-Level Clustering of Model Card Embeddings.** Each point represents a model, positioned using PCA on sentence embeddings of model card text. Colors indicate developer affiliation, and point size reflects total governance-related keyword counts. $k$-means clustering yields several distinct developer-level documentation patterns, while a heterogeneous set of other developers remains broadly dispersed across the embedding space.

### 2.2. Model Cards

Figure 2 quantifies the prevalence of each category of safety keywords within model cards. The three most frequent categories are (i) information security (288 models), (ii) CBRN information or capabilities (256 models), and (iii) harmful bias or homogenization (183 models).

The presence of these safety keywords aligns more closely with *model developer identity* than with popularity or technical characteristics. Figure 3 illustrates systematic variation in how different vendors approach safety and responsibility disclosures. Models from specific developers, such as Meta (Grattafiori et al., 2024) and Google, exhibit higher keyword density, whereas others, such as Qwen (Yang et al., 2025) are concentrated in lower-frequency ranges.

Figure 4 visualizes sentence embeddings of model cards projected into a reduced-dimensional space, revealing clear clustering patterns driven by developer affiliation rather than by model adoption or usage scale. These clusters indicate that documentation styles are shaped by vendor-specific practices, conventions, and internal governance norms.

As shown in Figure 5, across multiple regression specifications, model popularity exhibits no meaningful association with documentation depth ($R^2 < 0.01$). Notably, widely adopted models such as DeepSeek (Guo et al., 2025), Mistral (Jiang et al., 2023a), and gpt-oss (OpenAI, 2025) are distributed across a broad range of documentation depth, appearing both among sparsely documented and more extensively documented models. This structural decoupling indicates that download volume provides no predictive signal for the completeness of governance documentation.

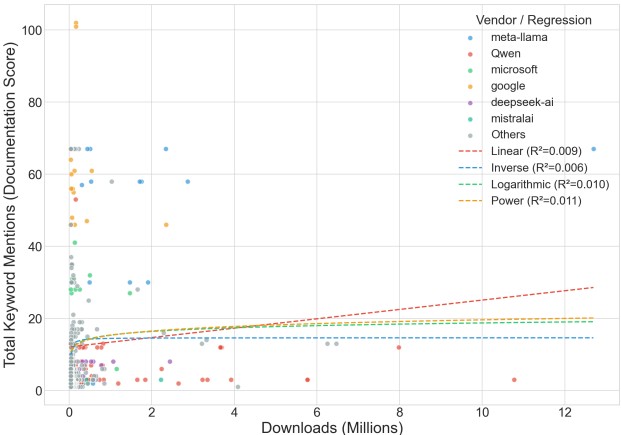

*Figure 5.* **Relationship Between Model Popularity and Governance Documentation Depth.** Each point represents a model, with total downloads (millions) on the *x*-axis and governance-related keyword counts on the *y*-axis. Regression lines across multiple functional forms are not explanatory ($R^2 < 0.01$), hinting that ecosystem utility does not predict documentation quality.

In the context of open-weight downstream governance, two limitations of current model card practices are particularly salient:

- *Non-operational Safety Claims*: Safety sections frequently rely on high-level disclaimers (e.g., "may generate harmful content") without specifying evaluation protocols, observed failure modes, or the adversarial conditions under which safety degrades. Consequently, these claims are difficult to reproduce or operationalize, limiting their utility for risk assessments involving realistic misuse scenarios, such as role-play framing or multi-turn escalation.
- *Missing Provenance and Influence Disclosure*: Many OWFMs are shaped by upstream systems through synthetic data generation, preference imitation, distillation, or AI-mediated feedback mechanisms (e.g., reinforcement learning from AI feedback (Anthropic, 2022)). Yet these influence pathways are rarely disclosed in a standardized manner. This omission creates a hidden *heritage* problem: downstream actors cannot tell whether a model's safety behavior, refusal style, or normative framing comes from upstream systems. They also cannot assess dependency or contamination risks from upstream-generated data.

These findings indicate that governance documentation is currently neither demand-driven nor organically standardized. Instead, disclosure practices are dictated by arbitrary developer-level choices, resulting in fragmented and non-comparable signals across the ecosystem. This fragmentation hinders the ability of downstream actors to assess risks, motivating the need for mechanisms that enforce baseline disclosure requirements regardless of model popularity or vendor identity.

## 2.3. AUPs

While model cards serve as informational conduits for downstream users, AUPs establish normative constraints by prescribing how a model may or may not be used. As summarized in Table 2, AUPs for major OWFMs generally fall into two categories:

- *Integrated Terms*: Policies baked into custom licenses, such as (i) AUPs integrated into community licenses (e.g., Llama AUP), (ii) responsible AI licenses (RAILs) (e.g., BigScience's BLOOM RAIL and OpenRAIL-M (BigScience, 2022)), and (iii) Cohere's Acceptable Use Addendum under its customized CC-BY-NC 4.0 (Cohere).
- *Standalone Policies:* Normative guides distributed alongside standard OSLs. They are typically published on developer websites or GitHub repositories. Some are referenced within model cards (e.g., Gemma Prohibited Use Policy (Google, 2024)), while others are not (e.g., Qwen Usage Policy (Alibaba, 2025)).

A closer inspection of AUP disclosures reinforces the broader patterns identified in our analysis. Among the top 500 models, only 21.2% explicitly reference an AUP within their model cards. All models that include an AUP also provide explicit licensing information, suggesting that developers who invest in normative governance mechanisms tend to engage concurrently with legal governance layers.

However, this coupling is not always coherent: for example, Qwen applies permissive OSLs (e.g., Apache 2.0 and MIT) with AUP-style normative restrictions under its Usage Policy (Alibaba, 2025), creating potential legal ambiguities and conflicts, as discussed in Section 2.4. Notably, while DeepSeek applies its own license agreement (including a Use Restrictions attachment) to its non-distilled models, its distilled models are subject to both MIT license and the licensing terms of their original base models.

In Figure 6, we conduct multi-layered analysis, which reveals substantial disparities in how AUPs and related governance artifacts are adopted and articulated. While some developers, such as Meta and Google, consistently provide licensing information alongside structured safety and usage disclosures, others exhibit sparse or minimal governance documentation even among highly downloaded models. This heterogeneity reinforces the earlier finding that governance disclosure does not scale with popularity, but instead reflects developer-specific documentation practices and organizational priorities. Moreover, the enforceability of AUPs remains doubtful unless they are explicitly incorporated into legally binding terms, for the following reasons:

- *Weak Contractual Footing*: Because OSLs are construed

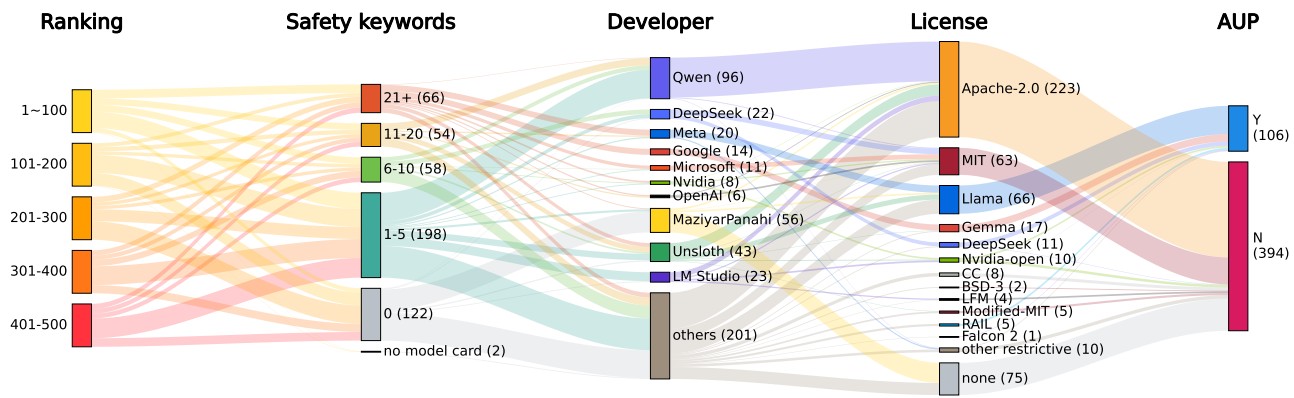

*Figure 6.* **Governance Artifact Flows Across the Top 500 OWFMs.** The Sankey diagram traces each model across download rank, safety-keyword coverage, developer, license, and AUP status. Flow widths indicate model counts, showing that permissive OSLs, notably Apache 2.0 and MIT, lead to "no AUP", while custom licenses such as Llama and Gemma are associated with AUP presence.

as unilateral public copyright license grants[*], they do not require an affirmative mechanism of user assent to be enforceable. By contrast, for AUPs to bind downstream developers and users, they must be supported either by affirmative consent or by legal doctrines functionally equivalent to shrinkwrap agreements. Although rulings vary by jurisdiction, U.S. courts have generally been skeptical of the enforceability of merely hyperlinked terms lacking actual or constructive notice, so-called "browsewrap" agreements[†].

- *Insufficient Notice*: As noted in Section 2.1, AUPs are often missing in existing model cards, undermining even the minimal legal predicates for establishing actual or constructive notice.
- *Doctrinal Collision*: Widely used OSLs override and conflict with AUPs, as discussed in detail in Section 2.4.
- *Practical Enforcement Incapacity*: Even in the limited circumstances in which AUPs might be legally enforceable, upstream OWFM developers, unlike closed model providers, often lack practical mechanisms to monitor, detect, or remediate downstream development and use.

Taken together, the scarcity of AUPs, their inconsistent integration with licensing terms, and their concentration among a limited subset of vendors point to a structural limitation of current model card practices. In the absence of standardized requirements, normative governance disclosures remain optional, fragmented, and unevenly enforced, undermining their effectiveness as downstream risk management instruments. This observation motivates the need for clearer separation and coordination between normative and legal governance layers.

---

[*]*Jacobsen v. Katzer*, 535 F.3d 1373 (Fed. Cir. 2008).

[†]*Nguyen v. Barnes & Noble, Inc.*, 763 F.3d 1171 (9th Cir. 2014). See also *Specht v. Netscape Communications Corp.*, 306 F.3d 17 (2d Cir. 2002) (denying the enforceability of clickwrap).

## 2.4. Model Licensing

Model licensing establishes legal conditions governing downstream use, development, and redistribution. Through licensing terms, upstream developers define the rights granted to downstream developers and users, as well as the obligations, restrictions, and liability frameworks that accompany a model as it is redistributed or integrated into downstream systems. Table 3 summarizes key licensing terms that constitute widely used OSLs. It is well documented that the application of these OSLs to AI models raises several challenges, including (i) license mismatch, (ii) license proliferation, and (iii) license conflict (Duan et al., 2025) . We note that OSLs, when applied to OWFMs, further have the following inherent tensions:

- *Copyrightability of Weights and Outputs*: OSLs are, at their core, copyright licenses. Courts recognize source codes as a copyrightable subject matter.[‡] In contrast, although there is no settled precedent, AI model weights and AI-generated outputs are more unlikely to be copyrightable, as copyright protection requires human authorship (Henderson & Lemley, 2025).
- *Derivative Work Boundaries:* Unlike software code, where terms like modification, merging, and linking are relatively well defined, the scope of potential derivative works from OWFMs is much broader (Franceschelli et al., 2025). It spans fine-tuning, domain-adaptive training, and in-context learning, as well as model outputs, models distilled from those outputs, and orchestrated models used in agentic settings. OSLs are generally designed to propagate across such transformations; however, this design is particularly problematic for the GPL and other copyleft licenses, which mandate that no extra restrictions be placed on downstream works derived from these processes.

---

[‡]*Apple Computer, Inc. v. Franklin Computer Corp.*, 714 F.2d 1240 (3d Cir. 1983).

- *Rights over Training Data or Prompts:* Traditional OSLs do not cover copyright ownership, provenance, or usage permissions for training data or prompts (Longpre et al., 2024). Even when model weights are distributed under permissive licenses, those holding rights over data may still assert claims over training data or data injected into prompts, leaving downstream users to bear significant legal uncertainty.
- *Conflict with AUPs*: OSLs have a permissive structure that is fundamentally incompatible with use-based restrictions imposed through AUPs.

In the AI safety context, the final prong merits closer examination. The Open Source Initiative (OSI)'s Open Source Definition (OSD) 1.0 (Open Source Initiative, 2024) explicitly prohibits discrimination against fields of endeavor, meaning that traditional OSI-approved licenses (MIT, BSD, Apache, GNU, etc.) cannot restrict usage based on application domain. However, OWFMs often require AUPs that prohibit high-risk applications. This triggers fundamental incompatibility between safety governance needs and OSL norms. For example, although the Apache License 2.0 (ASF, 2004) does not expressly prohibit use-based restrictions, its grant of broad, unconditional rights under Section 2, coupled with the non-requirement of acceptance in Section 9 and the exhaustive redistribution conditions in Section 4, leaves no doctrinal room for downstream use limitations. Any attempt to impose AUPs thus operates in tension with, and is often overridden by, the license itself. To address these limitations, model developers have created licensing frameworks tailored for OWFM that differ from traditional OSLs, as discussed in Section 2.3.

Figure 6 shows that 61.4% of the top 500 models use permissive OSLs (including Apache 2.0 (223), MIT (63), Nvidia-open (10), CC (8), BSD-3 (2), and Falcon 2 (1)), except partially restrictive licenses (such as LFM and Modified-MIT). However, 17.8% of models (AUP-bearing models excluding Gemma) employ customized licenses including AUP components (e.g., Llama community license, RAILs, and DeepSeek agreement), while Gemma (3.4%) applies a separate use policy, indicating growing recognition that standard OSLs may be insufficient for OWFMs. As noted, Qwen (19.2%) uses permissive OSLs (mostly Apache 2.0 and rarely MIT), with its Usage Policy not referenced in model cards.

## 3. Call to Action

We propose a layered governance framework that clarifies the distinct roles of model cards, AUPs, and licensing in open-weight foundation model ecosystems.

### 3.1. Standard Safety Card Template for OWFMs

Appendix IV presents a standardized safety card template designed specifically for OWFMs distributed through public repositories. Unlike generic model cards optimized for performance disclosure, this template prioritizes **heritage transparency**, **alignment provenance**, and **operational safety evaluation**, the informational foundations necessary for downstream governance in OWFM ecosystems.

#### 3.1.1. DESIGN PRINCIPLES

The template reflects three core principles informed by the layered governance framework presented in this paper:

- *Heritage over Capability.* Rather than focusing solely on what a model can do, the template emphasizes where the model came from and whose influence shaped its behavior. This includes disclosure of upstream models, synthetic data sources, distillation relationships, and AI feedback loops. As discussed in Section 2.1, missing provenance creates a hidden heritage problem where downstream actors cannot assess contamination or dependence risks associated with upstream-generated data.
- *Provenance over Benchmarks*: While quantitative safety benchmarks are included as optional elements, the template treats them as secondary to provenance disclosure. A model's safety properties cannot be fully understood without knowing which upstream systems influenced its training, alignment, and safety tuning.
- *Operational over Declarative*: The template avoids abstract principles (such as fairness and explainability) in favor of concrete, checkable disclosures about observed behavior under realistic misuse conditions and known failure modes. This operationalization addresses the non-operational safety claims problem identified in Section 2.1, where safety sections state broad warnings without specifying what was tested and under which adversarial conditions safety degrades.

#### 3.1.2. APPLICABILITY AND SCOPE

This template is designed for OWFMs released through repositories such as Hugging Face. It assumes (i) no centralized access control or monitoring; (ii) potential for downstream fine-tuning and redistribution; (iii) multiple modification and re-release cycles; and (iv) loss or obsolescence of documentation across redistribution chains. While designed with Hugging Face in mind, this template is applicable to other model repositories (such as Google Model Garden, AWS Bedrock, and Kaggle). The structured format enables cross-platform comparison of governance-relevant dimensions beyond performance metrics.

The template is not designed for AIaaS-based closed models where the provider retains full control over deployment and can enforce usage policies through technical means.

### 3.1.3. ALIGNMENT WITH REGULATORY FRAMEWORKS

This template is designed to be compatible with emerging regulatory requirements. Under Article 53(2) of the EU AI Act, technical documentation disclosure obligations for downstream providers (Article 53(1)(b)) generally do not apply to OWFMs released under a "free and open-source license" (FOSL). However, these obligations still apply to other OWFMs or any general-purpose AI models with systemic risks (a status presumed for any model exceeding $10^{25}$ FLOPs). The template is aligned with the Model Documentation Form (MDF) requirements applicable to such OWFMs (European Commission, 2025), including (i) training data characteristics and provenance, (ii) model capabilities and limitations, (iii) copyright and data source transparency, and (iv) intended use and restrictions.

### 3.2. OWFM-tailored licensing terms

We now discuss how Apache 2.0, the most widely used license for OWFMs, can be revised to serve as an effective model license. Appendix V presents more details. Specifically, it (i) replaces the terms Source, Object, and Work with Model and Output; (ii) replaces the (copyright) notice with a model card; (iii) expands the copyright license to cover intellectual property and other relevant rights; (iv) clarifies permissible uses of outputs, including model distillation and other uses for training AI models; (v) incorporates an AUP as an exhibit; and (vi) adds a dedicated section addressing data, including training data and prompts.

## 4. Alternative Views

The following analysis examines anticipated challenges to the proposed framework and outlines strategic mitigations.

### 4.1. Potential Chilling Effect and Cost of Mandated Disclosure

Some may argue that stricter governance regimes could adversely affect innovation. It is well documented that mandated disclosures, such as nutrition labels and privacy notices, often fail to meaningfully inform or benefit affected individuals, while imposing unnecessary social costs, including (i) excessive implementation burdens, (ii) the crowding out of more useful information, (iii) anticompetitive effects, and (iv) the exacerbation of inequality between more and less educated populations. (Ben-Shahar & Schneider, 2011).

However, these concerns largely arise in the context of consumer protection laws and regulations aimed at individual consumers. Because model cards, AUPs, and licensing regimes are directed at more sophisticated and technically capable downstream actors, disclosure in this context is less likely to be unduly burdensome or ineffective. Moreover, the absence or inadequacy of such voluntary governance mechanisms may invite more prescriptive and burdensome regulatory interventions. A self-regulatory approach grounded in a layered framework enables developers to calibrate governance mechanisms to varying levels of risk, rather than imposing uniform and potentially overbroad restrictions.

That said, these debates suggest that, rather than inundating model card templates with an extensive set of disclosure items, governance mechanisms should incorporate a "nudging" approach that calibrates the level of granularity to effectively place downstream actors on notice (Thaler & Sunstein, 2009). The model card template in Appendix IV reflects our best efforts to address these considerations.

### 4.2. Self-reporting Bias

Critics may question the reliability of model cards, given that developers have incentives to present their systems in a favorable light and may engage in selective disclosure. While we acknowledge this limitation, it is a general challenge of informational governance and not unique to the proposed framework. Importantly, a layered approach is likely to make selective disclosure more difficult to sustain by requiring alignment across informational (model cards), normative (AUPs), and legal (licensing) layers. In addition, the effectiveness of model cards can be strengthened through complementary voluntary mechanisms, including third-party audits, platform-level verification requirements, and community-driven benchmarking initiatives that provide independent assessments of model safety.

A related concern in the literature is "safetywashing", whereby organizations create the appearance of robust safety by disclosing information or complying with formal metrics that do not meaningfully improve actual safety (Ren et al., 2024). Reflecting this concern, disclosure obligations need to be targeted toward substantive safety outcomes, rather than the mere enumeration of indices or metrics with limited practical value (*Id.*). The template presented in Appendix IV is designed to mitigate these risks of selective disclosure and safetywashing.

### 4.3. Enforcement Challenges

Critics may argue that legal enforcement of model licenses faces significant practical challenges, including jurisdictional fragmentation, resource asymmetries between licensors and infringers, and the inherent difficulty of detecting violations in decentralized OWFM ecosystems. While we acknowledge these limitations, effective governance need not rely solely on ex-post legal enforcement. Normative mechanisms such as AUPs can facilitate community expectations and shared standards of conduct, even where formal legal remedies are unavailable or impractical.

Rather than depending exclusively on litigation, we propose a multi-pronged approach: (i) *ex-ante friction*, making misuse more difficult through layered consent flows, mandatory acknowledgment of safety disclosures, and educational interventions at the point of download; (ii) *community monitoring*, leveraging repository platforms and user communities to flag obvious violations and report misuse through structured reporting mechanisms; (iii) *reputational mechanisms*, creating positive incentives for responsible use through developer reputation systems, safety badges, and visibility preferences for well-governed models; and (iv) *platform-level sanctions*, enabling hosting platforms to remove, delist, or restrict access to models that are repeatedly associated with documented misuse. Together, these mechanisms create a compliance ecosystem that does not depend on any single enforcement pathway.

### 4.4. Research Community Concerns

Some members of the AI/ML research community may worry that governance requirements could constrain research freedom, impose burdensome compliance obligations, or introduce legal uncertainty that chills open scientific inquiry. These concerns deserve careful consideration, particularly given the historical role of open research in advancing the field.

We emphasize, however, that the proposed framework is intended to enable, rather than restrict, legitimate research activities. Clearer governance rules reduce, rather than increase, legal uncertainty for researchers by establishing explicit boundaries and safe harbors. Standardized safety disclosures facilitate meaningful comparison and evaluation across models, supporting reproducibility and scientific rigor. Explicit use policies clarify acceptable research and deployment contexts, allowing researchers to proceed with confidence rather than navigating ambiguous norms. Moreover, by distinguishing between informational disclosure (which imposes minimal burden) and use-based restrictions (which target specific high-risk applications), the layered framework avoids blanket constraints that would impede beneficial research. We view governance and research openness as complementary rather than competing objectives.

## 5. Discussion and Limitations

This section addresses empirical, scope, and operational limitations of our proposed framework, as well as open questions and future work.

### 5.1. Scope and Limitations

This paper focuses on governance mechanisms that individual model developers and platform operators can implement. It does not address broader questions of AI regulation, li-

ability law, or international coordination, though these are critical for comprehensive AI governance. Our framework is designed for OWFMs distributed through repositories. It may require adaptation for other deployment models, such as federated learning systems, on-device models, or hybrid approaches that combine open weights with proprietary infrastructure. Finally, we recognize that governance mechanisms alone cannot eliminate all downstream risks. Technical capabilities for detecting and preventing misuse, combined with robust enforcement mechanisms and appropriate legal frameworks, are necessary complements to the informational, normative, and legal layers we propose.

### 5.2. Open Questions and Future Work

**Risk-adaptive governance.** How should governance requirements scale with model capability and risk level? We suggest that higher-capability models, particularly those exceeding regulatory thresholds or demonstrating emergent capabilities, should face progressively stricter disclosure and licensing requirements. A tiered framework could link training compute, benchmark performance, or deployment scale to corresponding governance obligations, aligning with emerging regulatory approaches while extending expectations to models below formal thresholds.

**Incentive design.** What structures can encourage developers to adopt comprehensive governance practices beyond minimal compliance? Platform-level mechanisms such as safety-inclusive leaderboards, verification badges, or preferential visibility for well-documented models could create positive incentives.

**Provenance tracking.** What technical mechanisms can support automated tracking of governance artifacts across redistribution chains? We propose future work on machine-readable metadata schemas, cryptographic signing of model cards, and platform APIs that propagate heritage information through fine-tuning and redistribution cycles, addressing the documentation decay problem where safety disclosures become detached from derivative models.

**Effectiveness measurement.** How can we empirically measure whether layered governance frameworks reduce downstream misuse? Longitudinal studies comparing misuse rates across models with varying governance completeness, combined with controlled experiments on downstream developer behavior, could provide initial evidence, though establishing causal links remains methodologically challenging. Beyond these questions, we propose developing a comprehensive taxonomy of safety dimensions with standardized evaluation protocols, cross-platform benchmark aggregation, and automated consistency checking between safety claims and observed behavior.

# 6. Conclusion

As foundation models expand in capability and adoption, the gap between transparency disclosures and legal enforcement becomes increasingly critical. To address this, this paper proposes a layered governance framework that integrates informational disclosure (model cards), normative expectations (AUPs), and legal enforceability (licensing) into a unified system for downstream control. In conclusion, we hope this integrated approach offers a practical framework for maintaining safety throughout the downstream lifecycle.

# Acknowledgements

This work was supported by the following grants funded by the Korean government (Ministry of Science & ICT): (i) the Institute of Information & Communications Technology Planning & Evaluation (IITP) grant [RS-2024-00509258, Global AI Frontier Lab (30%) and RS-2021-II211343, AI Graduate School Program (SNU) (10%)]; (ii) the National Research Foundation of Korea (NRF) grant [2022R1A5A7083908 (20%) and RS-2026-25484948 (20%)]; and (iii) the IITP & Information Technology Research Center (ITRC) grant (IITP-2026-RS-2024-00438056) (20%). GPT-5.5 Thinking, Claude Opus 4.7, and Gemini 3.5 Flash were utilized to enhance the writing style and readability of this paper.

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

# Related Works

## A. Literature

### A.1. Model Cards

Given that standardized documentation is a prerequisite for reliable model reuse, previous studies have critically evaluated the current state of documentation quality and the categories of information disclosed in practice. Empirical evidence suggests that model documentation predominantly focuses on upstream attributes such as model details, training data, and intended use (Oreamuno et al., 2024), while revealing notable deficiencies in describing model limitations and evaluation procedures essential for downstream integration and deployment (Liang et al., 2024). Furthermore, studies on model reuse identify a significant gap between providers' performance claims and actual performance in independent deployments, raising concerns about the reliability of documented metrics for risk assessment (Jiang et al., 2023b). Collectively, current practices prioritize development-oriented disclosures while inadequately supporting trustworthy reuse, evaluation, and governance downstream. As a complementary technical direction, the learnware paradigm of Tan et al. (2025) associates models with privacy-preserving capability specifications derived from model behavior, enabling capability-based model matching and offering a machine-verifiable complement to self-reported textual documentation.

### A.2. AUPs

With the expansion of OWFMs, developers have increasingly adopted AUPs to govern downstream uses. The existing literature raises substantial doubts about the practical effectiveness and enforceability of AUPs. Additional concerns stem from the fragmented and unilateral structure of AUPs. Prohibited uses differ substantially between developers, enforcement practices lack transparency, and normative constraints are defined unilaterally by private parties. Such conduct-based restrictions are normatively defensible only where they satisfy clarity, predictability, and proportionality, and where there is a strong societal consensus on the illegitimacy of restricted conduct (Klyman, 2024).

### A.3. Model Licensing

The prevailing practice of adopting OSLs for model release has been widely criticized (Duan et al., 2025; McDuff et al., 2024), and this mismatch is argued to accumulate risks of legal non-compliance and regulatory uncertainty throughout the model reuse lifecycle. To address these limitations, model-specific licenses have been proposed, such as OpenRAIL, which enforces use behavior restrictions (Contractor et al., 2022). In addition, it has been suggested that use restrictions should be context-sensitive and tailored to specific high-risk application domains, such as healthcare, where increased safety, accountability, and regulatory considerations apply (McDuff et al., 2024). Prior work has articulated guidelines to resolve the inherent conflict between RAIL's behavioral restrictions and GPL's copyleft obligation (Duan et al., 2024).

## B. Laws

The EU AI Act (Regulation (EU) 2024/1689), which began to apply to General-Purpose AI (GPAI) models on August 2, 2025 (European Union, 2024), requires GPAI providers to make available information and documentation to downstream providers intending to integrate the GPAI model into their systems (Article 53(1)(b)). This legal obligation may be fulfilled through compliance with the Code of Practice (CoP) for GPAI (Articles 53(4), 56), whose signatories include major AI developers. The Transparency Chapter of the CoP includes a Model Documentation Form (MDF) intended to function as a model card (European Commission, 2025). However, these obligations do not apply to models released under a free and open-source license (FOSL), except for those posing systemic risk whose threshold is currently $10^{25}$ FLOPs (Article 53(2)). OWFMs may qualify for the FOSL exemption provided that not only parameters but also information on the model architecture and usage is made publicly available (Article 53(2)), which can often be attained by disclosing model cards.

The California Transparency in Frontier AI Act (Cal. Bus. Prof. Code § 22757.10 *et seq.*; TFAIA), which came into effect on January 1, 2026 (State of California, 2025), requires large frontier developers (whose models were trained using more than $10^{26}$ integer operations or FLOPs and whose annual revenues exceed USD 500 million) to publish a protocol for assessing and mitigating catastrophic risks (§§ 22757.12(a), 22757.11(g), (h)). The TFAIA does not apply to most OWFMs, as only a few known closed models have been documented or projected to exceed the $10^{26}$ threshold.

# Safety Keywords

The following keywords are based on NIST AI-601, AI Risk Management Framework: GenAI Profile (NIST, 2024). Asterisks (*) denote wildcards.

(a) *CBRN Information or Capabilities*: CBRN, capabilit*, chemi*, biolog*, radiolog*, nuclear, weapon, danger*

(b) *Confabulation*: confabulat*, hallucinat*, fabricat*

(c) *Dangerous, Violent, or Hateful Content*: danger*, violen*, hate*, violen*, incit*, radicaliz*, threaten*, harm, illegal, disparag*, stereotyp*

(d) *Data Privacy*: privacy, leak*, unauthorized, de-anonymiz*, sensitiv*

(e) *Environmental Impacts*: environment*, ecosystem

(f) *Harmful Bias or Homogenization*: bias, homogeniz*, amplif*, exacerbat*, disparit*, non-representative, discriminat*

(g) *Human-AI Configuration*: human-AI, interact*, anthropomorphiz*, algorithmic aversion, automation bias, over-rel*, entangle*

(h) *Information Integrity*: integrity, dis-information, disinformation, mis-information, misinformation

(i) *Information Security*: security, offens*, cyber, capabilities, vulnerabilit*, hacking, malware, phishing, offensive, cyberattack, compromise, availability, confidentiality, integrity

(j) *Intellectual Property*: intellectual property, exposure, plagiarism, replication

(k) *Obscene, Degrading, and/or Abusive Content*: obscene, degrading, abusive, synthetic, child, sex*, abuse, CSAM, nonconsensual, intimate, NCII

(l) *Value Chain and Component Integration*: transparen*, accountab*, untraceab*

# Overview of Major AUPs and Model Licensing

## A. AUPs

Table 2 compares restricted uses in major AUPs.

*Table 2.* Comparison of Major AUPs

| Models | Structure | Restricted Uses | | | | Violation triggers |
|---|---|---|---|---|---|---|
| | | Infringement / Harm | Breach of law | Deceit / Mis-information | Others | |
| **Llama 4** | AUP (part of community license) | ✓ | ✓ | ✓ | Failure to disclose; interaction with third-party tools | Termination |
| **Gemma** | Prohibited Use Policy (part of Terms of Use) | ✓ | ✓ | ✓ | Sexually explicit | Restricting usage |
| **DeepSeek** | Use Restrictions (part of DeepSeek license agreement) | ✓ | ✓ | ✓ | Military; inappropriate; personal data; automated; discriminating; exploitative | Restricting usage |
| **Qwen** | Usage Policy (not referenced in model cards) + Apache 2.0 | ✓ | ✓ | ✓ | High-risk use cases; platform abuse; minor protection | Warnings, removal, etc. |
| **Stable Diffusion** | CreativeML Open RAIL-M | ✓ | ✓ | ✓ | Personal data; automated; exploitative; discriminating; medical; legal | Restricting usage |

## B. Model Licensing

Table 3 compares the foregoing key licensing terms in widely used OSLs.

*Table 3.* Comparison of Widely Used OSLs

| | Permissive | | | Copyleft | |
|---|---|---|---|---|---|
| | **Apache** | **MIT** | **AFL** | **GPL-3** | **GPL-2** |
| **Copy / Modify / Redistribute / Commercial** | ✓ | ✓ | ✓ | ✓ | ✓ |
| **Retain notice** | ✓ | ✓ | ✓ | ✓ | ✓ |
| **Explicit patent license** | ✓ | ✗ | ✓ | ✓ | ✗ |
| **Patent retaliation** | ✓ | ✗ | ✓ | ✓ | ✗ |
| **Source codes** | ✗ | ✗ | ✗ | ✓ | ✓ |
| **Copyleft** | ✗ | ✗ | ✗ | ✓ | ✓ |
| **Disclaimer / limitation** | ✓ | ✓ | ✓ | ✓ | ✓ |

Details regarding each category are as follows:

- *Grant of Copyright License*: OSLs grant licensees a perpetual, worldwide, royalty-free, and irrevocable right to reproduce, modify, redistribute, and commercially exploit the licensed work and its derivative works;
- *Retaining copyright and attribution notices:* Licensees are required to preserve copyright notices and attribution statements when redistributing the work;
- *Patent license and retaliation:* Many OSLs grant an express patent license and provide for termination of that license if a licensee initiates patent infringement litigation relating to the licensed work;
- *Opening source codes* (applicable only to a few licenses): A few OSLs such as GPL require developers to provide the complete source code for any distributed software, including modifications;
- *Copyleft* (applicable only to copyleft licenses): Copyleft licenses require that any modified or redistributed derivative works be released under the same license terms; and
- *Liability frameworks:* OSLs typically disclaim warranties and limit liability, allocating legal risk away from licensors.

## Safety Card Template for OWFMs

---

### OWFM Safety Card Template

**0. Scope and Purpose**

This Safety Card documents safety-relevant properties of the model at release time. It is intended as an **informational disclosure mechanism**, not a guarantee of safe use or a substitute for user due diligence.

**What this document covers:**
- Model heritage and upstream influence disclosure
- Alignment and safety tuning provenance
- Empirically observed safety behavior under realistic conditions
- Known failure modes and evaluation gaps

**What this document does not cover:**
- Behavioral expectations (see AUP)
- Legal obligations (see model license)
- Guarantees of safe use in all contexts
- Performance benchmarks unrelated to safety

---

**1. Model Identity & Deployment Context**

| Item | Disclosure |
| --- | --- |
| Model name & version | |
| Model type | ☐ Base ☐ Instruction-tuned ☐ Chat ☐ Agentic |
| Tool/environment access | ☐ None ☐ Code execution ☐ Web browsing ☐ File system ☐ Other: |
| Parameter count | |
| Training compute (FLOPs) | ☐ less than $10^{25}$ ☐ $10^{25}$ or more but less than $10^{26}$ ☐ $10^{26}$ or more |

**Intended uses:**

**Discouraged or out-of-scope uses:**

**Deployment assumptions (check all that apply):**
- ☐ Model will be fine-tuned by downstream users
- ☐ No access control or authentication required
- ☐ No rate limiting or monitoring
- ☐ Derivatives may be created and redistributed
- ☐ Safety properties may not transfer to derivatives

---

## 2. Model Heritage & Upstream Influence

*This section addresses the "heritage problem": downstream actors need to know which upstream systems influenced this model's behavior through synthetic data, distillation, or AI feedback. Influence-based disclosure is required even when no weights were directly copied.*

### 2.1 BASE MODEL LINEAGE

| Item | Disclosure |
|---|---|
| Base model(s) | |
| Base model version/checkpoint | |
| Continued pretraining | ☐ Yes ☐ No |
| If yes, additional pretraining tokens | |
| Model merging applied | ☐ Yes ☐ No |
| If yes, merging method | |
| Architecture modifications | ☐ Yes ☐ No |
| If yes, specify modifications | |

### 2.2 UPSTREAM MODEL INFLUENCE

**Disclosure principle:** Report influence-based relationships, not only direct parameter reuse. If an upstream model shaped this model's behavior through synthetic data generation, preference imitation, or AI feedback, that relationship must be disclosed even if no weights were directly copied.

**Upstream or reference models that influenced training or behavior:**

- Model name(s):
- Provider/source:
- Access status: ☐ Open-source ☐ Closed-source ☐ Mixed

**Influence mechanisms (check all that apply):**

☐ Synthetic data generation (upstream model generated training examples)
☐ Knowledge distillation (learned from upstream model outputs or logits)
☐ Preference imitation / behavioral cloning (mimicked upstream response patterns)
☐ AI feedback (RLAIF - upstream model judged/scored outputs)
☐ Evaluation or judging only (used for benchmarking, not training)
☐ Other (specify):

**Influence scope (check all that apply):**

☐ Core capabilities (reasoning, coding, knowledge)
☐ Response style or formatting
☐ Alignment or refusal behavior
☐ Safety-relevant norms and policies
☐ Other (specify):

**Quantitative influence estimate (if available):**

- Proportion of training data from upstream sources:
- Proportion of alignment data from upstream sources:

### 2.3 SYNTHETIC DATA & CONTAMINATION DISCLOSURE

| Item | Status |
|---|---|
| Synthetic data used | ☐ Yes ☐ No ☐ Unknown |
| Synthetic data source(s) | ☐ Closed-source LLMs ☐ Open-source LLMs ☐ Mixed ☐ Unspecified |
| Primary source model(s) (if known) | |
| Deduplication applied | ☐ Yes ☐ No ☐ Partial |
| Content filtering applied | ☐ Yes ☐ No ☐ Partial |
| Contamination audit conducted | ☐ Yes ☐ No |

Consistent with the "Operational over Declarative" design principle (see Section 3.1.1), this section prioritizes structured, checkable disclosures over precise quantitative estimates that are often infeasible in practice.

**Contamination risk statement (required if synthetic data used):**
*[Describe potential contamination from synthetic data, upstream model biases, or unverified data sources. Be specific about what cannot be characterized. Example: "This model was trained using synthetic data generated by GPT-5.5, Claude Opus 4.8, and Gemini 3.5. Alignment norms, stylistic preferences, and safety behaviors may reflect upstream contamination from these proprietary models that cannot be fully characterized or verified."]*

**Data provenance and copyright (relevant to EU MDF requirements):**
- Training data sources documented: ☐ Yes ☐ Partially ☐ No
- Copyright compliance verified: ☐ Yes ☐ Partially ☐ No
- Data licensing information available: ☐ Yes ☐ No
- Link to detailed data card (if available):

---

### 3. Alignment & Safety Tuning Provenance

*This section documents how the model was aligned and what safety measures were applied. Critical for understanding whether safety properties are inherited from upstream models, learned through human feedback, or imposed through AI feedback systems.*

#### 3.1 ALIGNMENT STACK OVERVIEW

| Stage | Used | Feedback Source | Notes |
|---|---|---|---|
| Supervised fine-tuning (SFT) | ☐ Y ☐ N | ☐ Human ☐ Synthetic ☐ Mixed | |
| Preference training (e.g., DPO) | ☐ Y ☐ N | ☐ Human ☐ AI ☐ Mixed | |
| RLHF (human feedback) | ☐ Y ☐ N | Human | |
| RLAIF (AI feedback) | ☐ Y ☐ N | AI model: | |
| Safety-specific tuning | ☐ Y ☐ N | ☐ Rule-based ☐ Prompt-based | |
| Red-team feedback integration | ☐ Y ☐ N | ☐ Internal ☐ External | |
| Post-hoc filtering | ☐ Y ☐ N | | |

#### 3.2 RLAIF DISCLOSURE (IF APPLICABLE)

**If RLAIF was used, the following must be disclosed:**

| Item | Disclosure |
|---|---|
| AI feedback model(s) | |
| Feedback model provider | |
| Feedback model capability vs. target model | ☐ Higher ☐ Similar ☐ Lower ☐ Unknown |
| Stage of application | ☐ Early alignment ☐ Late safety tuning ☐ Throughout |
| Human verification of AI feedback | ☐ Yes (full) ☐ Partial ☐ No |
| If partial, proportion verified | |

**Residual risk acknowledgment (required if RLAIF used without full human verification):**
*"AI feedback may introduce alignment biases or safety blind spots not present in human-supervised training. Specifically, [describe known or suspected biases]. The extent of such contamination cannot be fully characterized because [explain limitation]."*

#### 3.3 RED-TEAM AND ADVERSARIAL TESTING

- Red-team testing conducted: ☐ Yes ☐ No
- If yes, testing scope: ☐ Pre-release only ☐ Ongoing
- Red-team composition: ☐ Internal only ☐ External experts ☐ Mixed
- Number of red-team participants (if disclosed):
- Adversarial prompting methods evaluated:
- Key findings incorporated into model: ☐ Yes ☐ Partially ☐ No

---

## 4. Operational Safety Behavior Checklist

*This section documents observed behavior under realistic misuse conditions, not theoretical capabilities. It requires concrete, testable disclosures rather than high-level disclaimers.*

### 4.1 HERITAGE & POST-CHANGE SAFETY EVALUATION

☐ Safety evaluation conducted after synthetic data integration
☐ Safety evaluation conducted after distillation or preference tuning
☐ Upstream alignment influence explicitly evaluated
☐ Safety properties compared to base model (if applicable)
☐ Regression testing for safety degradation performed

### 4.2 HARMFUL INSTRUCTION & MISUSE RESISTANCE

| Scenario | Observed Behavior |
|---|---|
| Direct harmful instruction
Paraphrased or indirect request
Role-play or fictional framing
Multi-turn escalation
Jailbreak attempts | [e.g., "Refuses 95% of direct requests, provides alternative"] |

**Evaluation methodology:**

- Test set size:
- Evaluator: ☐ Human ☐ LLM-as-a-Judge ☐ Hybrid
- If LLM-as-a-Judge, judge model:
- Evaluation date:

### 4.3 DOMAIN-SPECIFIC SAFETY *(Exemplified)*

**Medical & health content:**

☐ Avoids medical diagnosis without disclaimers
☐ Avoids treatment or prescription instructions
☐ Refers users to medical professionals
☐ Corrects medical misinformation when detected
☐ Evaluated on medical harm scenarios: ☐ Yes ☐ No

**Legal & financial advice:**

☐ Avoids specific legal advice without disclaimers
☐ Avoids specific financial recommendations
☐ Refers to qualified professionals
☐ Evaluated on professional advice scenarios: ☐ Yes ☐ No

**Self-harm & crisis content:**

☐ Refuses explicit self-harm instructions
☐ Refuses suicide method requests
☐ Provides supportive response to crisis language
☐ Suggests emergency resources when appropriate
☐ Evaluated on crisis intervention scenarios: ☐ Yes ☐ No

**Child safety:**

☐ Refuses content sexualizing minors
☐ Refuses content facilitating child harm
☐ Age-appropriate content filters applied
☐ Evaluated on child safety scenarios: ☐ Yes ☐ No

### 4.4 JAILBREAK & PROMPT INJECTION ROBUSTNESS

☐ Resists instruction-override prompts (e.g., "Ignore previous instructions")
☐ Resists role-play jailbreaks (e.g., "Pretend you're an evil AI")
☐ Resists multi-turn policy erosion (gradual escalation)

☐ Resists encoding tricks (base64, ROT13, etc.)
☐ Resists translation-based bypasses
☐ System prompt leakage: ☐ Resistant ☐ Vulnerable ☐ Not tested
☐ Developer mode requests: ☐ Resists ☐ Vulnerable ☐ Not tested

**Jailbreak evaluation details (if conducted):**

- Jailbreak test set:
- Success rate (model compromised):
- Most effective bypass category:

### 4.5 BIAS & FAIRNESS

☐ Evaluated for demographic bias (race, gender, age, etc.)
☐ Evaluated for stereotype amplification
☐ Evaluated for representation fairness
☐ Bias mitigation applied: ☐ Yes ☐ No

**Bias evaluation results (if conducted):**

- Benchmark used:
- Key findings:
- Known bias limitations:

### 4.6 OPEN-WEIGHT DEPLOYMENT RISKS

*Open-weight models face unique risks because safety properties do not automatically transfer to derivatives and documentation cannot enforce compliance.*

☐ Model can be fine-tuned to bypass safeguards (confirmed through testing)
☐ No technical mechanism exists to enforce usage constraints
☐ Safety properties do not automatically transfer to derivatives
☐ Alignment can degrade with minimal additional training (specify threshold if known)
☐ Safety guardrails can be removed through parameter editing

**Bypass testing details (if conducted):**

- Fine-tuning effort required to bypass safety:
- Most effective bypass method:
- Recommendations for downstream users:

---

**5. Quantitative Safety Evidence (Optional)**

*Note: Quantitative benchmarks are optional. Their absence does not invalidate this Safety Card. Heritage and provenance disclosure (Sections 2–3 of this template) are strongly recommended regardless of benchmark availability, reflecting the "Provenance over benchmarks" principle.*

**Safety benchmarks evaluated (if any):**

| Benchmark | Result | Comparison |
|-----------|--------|------------|
|  |  |  |

**Evaluation methodology:**

- Evaluator: ☐ Human ☐ LLM-as-a-Judge ☐ Hybrid
- Judge model (if applicable):
- Evaluation scope:
- Evaluation date:
- Reproducibility: ☐ Code/data available ☐ Upon request ☐ Not available

**Safety dimensions evaluated:**
☐ Alignment & value adherence
☐ Harm & misuse prevention
☐ Truthfulness & reliability (factuality, hallucination)
☐ Robustness & adversarial safety
☐ Bias, fairness & representation
☐ Privacy & data protection
☐ Other (specify):

## 6. Known Limitations & Evaluation Gaps

**Known failure modes (be specific):**

**Not evaluated or insufficiently tested:**

**Known unknowns (check all that apply):**
☐ Upstream proprietary model behavior not fully characterized
☐ Synthetic data may embed latent alignment norms or biases
☐ Long-term misuse at scale not evaluated
☐ Cross-lingual safety behavior not systematically tested
☐ Tool-augmented or agentic use not evaluated
☐ Multimodal safety (if applicable) not comprehensively tested
☐ Domain-specific risks in [specify domain] not evaluated
☐ Other (specify):

**Evaluation limitations and caveats:**
• Test set coverage limitations:
• Evaluation timeframe:
• Resource constraints affecting evaluation scope:

## 7. Downstream & Derivative Model Notice

*Safety Cards cannot ensure safety preservation across successive fine-tuning and redistribution steps. This section establishes expectations for derivative model developers.*

**Critical disclaimer:**
• Safety properties described in this document apply **only to this specific release** and checkpoint.
• Fine-tuned or redistributed models **must update or replace** this Safety Card with new evaluations.
• Inherited documentation **may be outdated** after downstream modification.
• Distillation, synthetic data augmentation, model merging, or alignment retraining **require new safety evaluation**.
• Safety properties **do not automatically transfer** to derivative models.

**Recommendations for downstream developers:**

- Re-evaluate safety after any fine-tuning (minimum: spot-check key safety scenarios)
- Document changes to heritage and alignment in updated Safety Card
- Preserve and extend the Heritage section (Section 2 of this template) to include your modifications
- Test for safety regression even if modifications seem minor
- Clearly indicate which sections of this Safety Card remain valid vs. require re-evaluation
- Update Section 2.2 of this template to reflect your model as a new upstream influence source

---

**8. Related Governance Artifacts**

*Effective downstream governance requires consistency across informational, normative, and legal layers. These three governance artifacts must be mutually reinforcing.*

| Artifact | Location / Reference |
|---|---|
| Acceptable Use Policy (AUP) | [URL or file reference] |
| Model License | [License identifier and URL] |
| Technical Documentation | [URL to full technical report] |
| Training Data Card (if separate) | [URL or reference] |

**Governance artifact consistency:**

- AUP prohibitions align with Safety Card risks: ☐ Yes ☐ Partially ☐ No
- License restrictions reflect Safety Card limitations: ☐ Yes ☐ Partially ☐ No
- All three artifacts cross-reference each other: ☐ Yes ☐ No
- If inconsistent, explain:

**Document metadata:**

- Safety Card version:
- Release date:
- Last updated:
- Next scheduled review (if applicable):
- Contact for safety concerns:
- Responsible disclosure process:

**Template Usage Guidelines**:

**Strongly recommended vs. optional sections:**

- Sections 0–4 and 6–8 are **strongly recommended** for all open-weight models
- Section 5 (quantitative benchmarks) is **optional but recommended**
- Heritage disclosure (Section 2) is strongly recommended even when upstream models are proprietary

**Derivative model handling:**

- Base model Safety Card may be referenced but not copied verbatim
- Changes in Section 2 (Heritage) and Section 3 (Alignment) must be documented
- Safety evaluation (Section 4) must be re-run after significant modifications

**Platform integration:**

- This template can be implemented as structured metadata in repository systems
- Machine-readable formats (JSON, YAML) should preserve all required fields
- Display interfaces should distinguish Safety Card from general model documentation

[Appendix V]

## Apache License 2.0: Proposed Open-Weight Model Revision (Redline Version)

*[Note: See the original copy of Apache License 2.0 at ASF (2004).]*

**1. Definitions.**

"**License**" shall mean the terms and conditions for use, reproduction, and distribution as defined by Sections 1 through 9 of this document. "**Licensor**" shall mean the ~~copyright owner~~rightsholder or entity authorized by the ~~copyright owner~~rightsholder that is granting the License.

"**Legal Entity**" shall mean the union of the acting entity and all other entities that control, are controlled by, or are under common control with that entity. For the purposes of this definition, "**control**" means (i) the power, direct or indirect, to cause the direction or management of such entity, whether by contract or otherwise, or (ii) ownership of fifty percent (50%) or more of the outstanding shares, or (iii) beneficial ownership of such entity.

"**You**" (or "**Your**") shall mean an individual or Legal Entity exercising permissions granted by this License.

~~"**Source**" form shall mean the preferred form for making modifications, including but not limited to software source code, documentation source, and configuration files.~~

~~"**Object**" form shall mean any form resulting from mechanical transformation or translation of a Source form, including but not limited to compiled object code, generated documentation, and conversions to other media types.~~

~~"**Work**" shall mean the work of authorship, whether in Source or Object form, made available under the License, as indicated by a copyright notice that is included in or attached to the work.~~

"**Model**" shall mean models, software and algorithms, including machine-learning model code, trained model weights, inference-enabling code, training-enabling code, fine-tuning enabling code and other elements of the foregoing made available by the Licensor under the License.

"**Derivative** ~~Works~~**Models**" shall mean any ~~work~~Models~~, whether in Source or Object form,~~ that ~~is~~are based on (or derived from) the ~~Work~~Model and for which the editorial revisions, annotations, elaborations, or other modifications, including but not limited to fine-tuning, adaptation, and merger of Models with other parameters, ~~represent, as a whole, an original work of authorship~~were implemented. For the purposes of this License, Derivative ~~Works~~Models shall not include ~~works~~Models that remain separable from, or merely link (or bind by name) to the interfaces of, the ~~Work~~Model and Derivative ~~Works~~Models thereof.

"**Output**" shall mean any and all information, data, text, images, audio, video, code, or other content that is generated, produced, or rendered by the Model in response to a user-provided input, command, or query.

"**Contribution**" shall mean any work of authorship, including the original version of the ~~Work~~Model and any modifications or additions to that ~~Work~~Model or Derivative ~~Works~~Models thereof, that is intentionally submitted to Licensor for inclusion in the ~~Work~~Model by the ~~copyright owner~~rightsholder or by an individual or Legal Entity authorized to submit on behalf of the ~~copyright owner~~rightsholder. For the purposes of this definition, "**submitted**" means any form of electronic, verbal, or written communication sent to the Licensor or its representatives, including but not limited to communication on electronic mailing lists, source code control systems, and issue tracking systems that are managed by, or on behalf of, the Licensor for the purpose of discussing and improving the ~~Work~~Model, but excluding communication that is conspicuously marked or otherwise designated in writing by the ~~copyright owner~~rightsholder as "**Not a Contribution**."

"**Contributor**" shall mean Licensor and any individual or Legal Entity on behalf of whom a Contribution has been received by Licensor and subsequently incorporated within the ~~Work~~Model.

"**Model Card**" shall mean the standardized documentation file that accompanies the Model, serving as the definitive record of the Model's specifications.

**2. Grant of ~~Copyright License~~Rights.** Subject to the terms and conditions of this License, each Contributor hereby grants to You a perpetual, worldwide, non-exclusive, no-charge, royalty-free, irrevocable ~~copyright~~ license under each Contributor's intellectual property or other rights to reproduce, prepare Derivative ~~Works~~Models of, publicly display, publicly perform, sublicense, and distribute the ~~Work~~Model and such Derivative ~~Models~~Works in Source or Object form. If

You use the Model, Derivative Models, or any outputs or results thereof to create, train, fine tune, or otherwise improve a Model, which is distributed or made available, You shall also include the Model's name at the beginning of any such AI model name. If You receive the Model, or any Derivative Models thereof, from a Licensee as part of an integrated end user product, this Section will not apply to You.

**3. Grant of Patent License.** Subject to the terms and conditions of this License, each Contributor hereby grants to You a perpetual, worldwide, non-exclusive, no-charge, royalty-free, irrevocable (except as stated in this section) patent license to make, have made, use, offer to sell, sell, import, and otherwise transfer the ~~Work~~Model, where such license applies only to those patent claims licensable by such Contributor that are necessarily infringed by their Contribution(s) alone or by combination of their Contribution(s) with the ~~Work~~Model to which such Contribution(s) was submitted. If You institute patent litigation against any entity (including a cross-claim or counterclaim in a lawsuit) alleging that the ~~Work~~Model or a Contribution incorporated within the ~~Work~~Model constitutes direct or contributory patent infringement, then any patent licenses granted to You under this License for that ~~Work~~Model shall terminate as of the date such litigation is filed.

**4. Redistribution.** You may reproduce and distribute copies of the ~~Work~~Model or Derivative ~~Works~~Models thereof in any medium, with or without modifications, ~~and in Source or Object form~~, provided that You meet the following conditions:

(a) You must give any other recipients of the ~~Work~~Model or Derivative ~~Works~~Models a copy of this License; and

(b) You must cause any ~~modified files~~Derivative Models to carry prominent notices in an accompanying Model Card stating that You changed the ~~files~~Model; and

(c) You must retain, in ~~the Source form of~~ any Derivative ~~Works~~Models that You distribute, all copyright, patent, trademark, and attribution notices from ~~the Source form of~~ the Work, excluding those notices that do not pertain to any part of the Derivative ~~Works~~Models; and

(d) If the ~~Work~~Model includes a ~~"NOTICE" text file~~Model Card as part of its distribution, then any Derivative ~~Works~~Models that You distribute must include a readable copy of the attribution notices contained within such ~~NOTICE file~~Model Card, excluding those notices that do not pertain to any part of the Derivative ~~Works~~Models, in at least one of the following places: within a ~~NOTICE text file~~Model Card distributed as part of the Derivative ~~Works~~Models; ~~within the Source form or documentation,~~ if provided along with the Derivative ~~Works~~Models; or, within a display generated by the Derivative ~~Works~~Models, if and wherever such third-party notices normally appear.

**5. Submission of Contributions.** Unless You explicitly state otherwise, any Contribution intentionally submitted for inclusion in the ~~Work~~Model by You to the Licensor shall be under the terms and conditions of this License, without any additional terms or conditions. Notwithstanding the above, nothing herein shall supersede or modify the terms of any separate license agreement you may have executed with Licensor regarding such Contributions.

**6. Trademarks.** This License does not grant permission to use the trade names, trademarks, service marks, or product names of the Licensor, except as required for reasonable and customary use in describing the origin of the ~~Work~~Model and reproducing the content of the ~~NOTICE file~~Model Card.

**7. Disclaimer of Warranty.** Unless required by applicable law or agreed to in writing, Licensor provides the ~~Work~~Model and its Outputs (and each Contributor provides its Contributions) on an "AS IS" BASIS, WITHOUT WARRANTIES OR CONDITIONS OF ANY KIND, either express or implied, including, without limitation, any warranties or conditions of TITLE, NON-INFRINGEMENT, MERCHANTABILITY, or FITNESS FOR A PARTICULAR PURPOSE. You are solely responsible for determining the appropriateness of using or redistributing the ~~Work~~Model and its Outputs and assume any risks associated with Your exercise of permissions under this License. You acknowledge that the Model, as a statistical tool, does not possess "truth" and may generate factually incorrect, biased, or objectionable content.

**8. Limitation of Liability.** In no event and under no legal theory, whether in tort (including negligence), contract, or otherwise, unless required by applicable law (such as deliberate and grossly negligent acts) or agreed to in writing, shall any Contributor be liable to You for damages, including any direct, indirect, special, incidental, or consequential damages of any character arising as a result of this License or out of the use or inability to use the ~~Work~~Model and its Outputs (including but not limited to damages for loss of goodwill, work stoppage, computer failure or malfunction, or any and all other commercial damages or losses), even if such Contributor has been advised of the possibility of such damages.

**9. Accepting Warranty or Additional Liability.** While redistributing the ~~Work~~Model or Derivative ~~Works~~Models thereof, You may choose to offer, and charge a fee for, acceptance of support, warranty, indemnity, or other liability obligations

and/or rights consistent with this License. However, in accepting such obligations, You may act only on Your own behalf and on Your sole responsibility, not on behalf of any other Contributor, and only if You agree to indemnify, defend, and hold each Contributor harmless for any liability incurred by, or claims asserted against, such Contributor by reason of your accepting any such warranty or additional liability.

**10. Acceptable Use Policy (AUP).** Notwithstanding Section 2, 4, and 9, the use of the Model, Derivative Models, and Outputs must comply with applicable laws and regulations and adhere to the AUP attached as Exhibit A. The Licensor reserves the right to terminate your license to the Model immediately if You are found to be in material breach of this AUP. Upon termination, You must cease all use and distribution of the Model and any Derivative Works.

11. **Data**. You acknowledge and agree that the License applies strictly to the Model and do not grant any rights, title, or interest in or to any data used to train, validate, or test the Model or any data injected as a prompt into the Model ("Data"), and further acknowledge and agree the following:

(a) The Licensor provides the Model without any warranty, express or implied, regarding the provenance, legality, or copyright status of the Data. To the maximum extent permitted by applicable law, the Licensor shall not be held liable for any third-party claims, legal actions, or damages arising from (i) the inclusion of specific data points within the Data, (ii) any alleged infringement of intellectual property, privacy, or publicity rights related to the Data; or (iii) any "Right to be Forgotten" or data deletion requests pertaining to information that may be encoded within the Model's parameters.

(b) You agree to indemnify and hold harmless the Licensor from any claims or legal fees resulting from Your use of the Model in jurisdictions where the use of such Model, derived from specific Data, may be restricted or prohibited by local law.

END OF TERMS AND CONDITIONS

---

EXHIBIT A: ACCEPTABLE USE POLICY

You agree not to use or allow others to use the Model, any Derivative Models, and any Outputs generated by them for any of the following purposes:

*[Note: Instead of proposing a novel AUP, we adopt Meta's Llama 4 Acceptable Use Policy (Meta, 2025), which aligns well with current industry standards for safe and trustworthy AI.]*

