# OpenReview forum: "Position: Current Model Cards Are Insufficient for Downstream Governance of Open-Weight Foundation Models"
_ICML.cc/2026/Position_Paper_Track — ICML 2026 Position Paper Track regular_

### Official Review · Reviewer_ctat · 2026-03-12

**Significance:** 3
**Argument Clarity:** 3
**Rating:** 4
**Confidence:** 3

**Questions:**

N/A

**Alternative Views Section:**

Yes

**Compliance With Llm Reviewing Policy A Conservative:**

Affirmed.

**Discussion Potential:**

3

**Paper Summary:**

The current paper puts forth the position that existing model cards for open-weight foundation models are insufficient for downstream governance. In particular, the authors look through 500 model cards on HuggingFace (the leading open weight model provider) and argue that effective governance is not achieved by the majority of model cards. Finally, they provide detailed guidance for transforming model cards, acceptable use policies, and licenses in order to create a more comprehensive governance framework.

**Position:**

Yes

**Position In Title:**

Yes

**Related Work:**

3

**Strengths And Weaknesses:**

1. Figure 1 is quite illustrative and depicts the ideal approach that the authors are putting forward in an easy to understand manner.
2. The fact that the authors went through 500 model cards is rather compelling, ensuring that their findings are grounded in what the actual state of the model card status quo appears to be.
3. the call to action properly provides a structured template for a proposed safety card, along with design principles and how well this proposed artifact aligns with existing regulatory frameworks. The attention to this will ensure that the operationalization of these safety cards is tractable.
4. the discussion around incentive design is welcome. In particular, given the widespread nature of the open weight foundation model landscape, it is refreshing to read discourse around what the various incentives are (and what they could be) so as to encourage the proper adoption of the proposed governance framework changes.

**Support:**

3

---

> ### Author Rebuttal · Authors · 2026-03-30
>
> We thank the reviewer for the careful reading and for the consistently positive assessment of our work.
>
> We are glad that Figure 1, the empirical analysis of 500 model cards, and the Safety Card Template were well received. We particularly appreciate the recognition of the incentive design discussion in Section 4.3. We agree that governance adoption in decentralized ecosystems is ultimately an incentive design problem, and we will clarify this practical adoption pathway more explicitly in the revised paper.
>
> We also emphasize that the Safety Card is designed as a **compliance-ready artifact**, aligned with emerging documentation expectations, thereby creating direct adoption incentives rather than relying solely on voluntary goodwill.

---

> > ### Author Rebuttal · Reviewer_ctat · 2026-04-06
> >
> > I have read the rebuttal and will maintain my score.

---

### Official Review · Reviewer_vJPe · 2026-03-13

**Significance:** 4
**Argument Clarity:** 3
**Rating:** 3
**Confidence:** 4

**Questions:**

- How can developers practically guarantee or even accurately estimate the proportion of synthetic data in their pretraining datasets?

- If we restrict the users from certain usage, then how open source the model is?

**Alternative Views Section:**

Yes

**Compliance With Llm Reviewing Policy A Conservative:**

Affirmed.

**Discussion Potential:**

4

**Final Justification:**

Thank you for your continued efforts to refine your stance. However, my main concerns remain unsolved.

The framework works against the open-source community and will cause friction in adoption. The proposed categorical data tracking is not strict and does not provide helpful information regarding test-set contamination and does not help us better understand the black-box models. Secondly, adding a burden of managing custom licenses and AUPs will put a friction on the open-weight community. This only benefits the closed-weight models which can avoid such burden.

Therefore, for these reasons I am still unconvinvced that this framework is practical and helpful to the open-weight community.

**Paper Summary:**

The authors argue that current AI documentation is inadequate for open-weight foundation models (OFWMs). They consider a broad aspect of AI governance by addressing the critical disconnect between informational (model cards), normative (AUPs), and legal (licenses) layers. To resolve this, the authors propose a unified, three-layered framework that legally binds usage restrictions to model licensing.

**Position:**

Yes

**Position In Title:**

Yes

**Related Work:**

2

**Strengths And Weaknesses:**

Strengths:

- The authors back their claims with a robust empirical analysis of the 500 open-weight models on HuggingFace.

- The authors propose a three-layered governance framework, which is highly structured and logical to follow.

- The authors argue that current licenses such as MIT or Apache 2.0 are insufficient for current AI models due to their high risk use cases.


Weaknesses:

- Many datasets now consist of synthetic (distilled) data from many different models, then it is really hard to trace the influence of those data on the model. The best solution would be to release the data with their sources.

- The authors assume the models are mainly hosted on the HuggingFace, but there are many different ways the models can be shared and it would be hard to enforce any policies.

- It seems that this Safety Card requires a lot of efforts and resources, which only large labs can afford to fill out necessary information, which small labs might lack of.

**Support:**

3

---

> ### Author Rebuttal · Authors · 2026-03-30
>
> We thank the reviewer for the thoughtful and constructive feedback. We appreciate the recognition of the empirical grounding and the clarity of our three-layer governance framework. Below, we provide a detailed response addressing each of the weaknesses [W#] and questions [Q#] raised.
>
> ---
>
> **[Q1]** We agree that accurately quantifying the proportion of synthetic data in web-scale pretraining corpora is fundamentally challenging, and the reviewer is correct that releasing training data with their sources is the ideal solution. Rather than assuming precise measurement, our intention is to encourage **reasonable transparency signals** rather than strict guarantees.
>
> In light of this, we will revise the Safety Card Template (Appendix IV, Section 2.3) to **relax the requirement for precise proportion estimates**, replacing mandatory quantitative fields with **coarse-grained categorical disclosures**: (i) whether synthetic data was used (Yes / No / Unknown), (ii) the category of source (closed-source LLMs / open-source LLMs / mixed / unspecified), and (iii)
> a required *Contamination Risk Statement* that explicitly acknowledges the limits of characterization. This revision shifts the standard from false precision to structured transparency, consistent with the "Operational over Declarative" principle (Section 3.1.1). We thank the reviewer for this concrete and constructive suggestion.
>
> ---
>
> **[Q2]** While the OSI definition prohibits restrictions on "fields of endeavor," the unique risks posed by foundation models necessitate an evolution in licensing. In such cases, the model is better characterized as an **Open Weights model** rather than Open Source under OSI standards. This distinction is intentional: while we embrace the transparency and collaborative spirit of Open Source, we prioritize safety-by-design. By implementing usage restrictions, we are not limiting openness for the sake of profit, but for the sake of public safety, creating a new class of license that balances technical accessibility with ethical accountability.
>
> ---
>
> **[W1]** As noted in our response to [Q1] above, rather than requiring full traceability, our framework emphasizes **disclosure of data generation processes and known upstream dependencies**, providing partial but actionable transparency. We will revise the manuscript to clarify that the framework promotes **best-effort provenance signaling** under realistic constraints rather than assuming perfect traceability.
>
> ---
>
> **[W2]** We agree that models are distributed through a variety of channels beyond Hugging Face. Our use of Hugging Face data is intended as a **representative example of current model distribution ecosystems**, rather than a limiting assumption. Hugging Face serves as the primary public distribution channel for many major OWFM. This can be supported by:
> - Jiang, Wenxin, et al. "An empirical study of pre-trained model reuse in the hugging face deep learning model registry." 2023 IEEE/ACM 45th International Conference on Software Engineering (ICSE). IEEE, 2023.
> - Laufer, Benjamin, Hamidah Oderinwale, and Jon Kleinberg. "Anatomy of a machine learning ecosystem: 2 million models on hugging face." arXiv preprint arXiv:2508.06811 (2025).
>
> However, we acknowledge this as a scope limitation in Section 5.1, and note in Section 3.1.2 that the Safety Card Template is applicable across other repositories beyond Hugging Face, including Google Model Garden, AWS Bedrock, and Kaggle.
>
> - https://cloud.google.com/model-garden
> - https://docs.aws.amazon.com/bedrock/latest/userguide/what-is-bedrock.html
> - https://www.kaggle.com/
>
> For models distributed outside major repositories, the OWFM-tailored license (Appendix V, Section 10) provides a contractual mechanism that travels with the model regardless of distribution channel, and the multi-pronged compliance ecosystem described in Section 4.3, including community monitoring, reputational mechanisms, and platform-level sanctions, does not depend on centralized repository enforcement. We will clarify how the framework generalizes to other distribution settings in the revised paper.
>
> ---
>
> **[W3]** We tried to address this important concern in Section 4.1 of the paper. As noted there, the absence or inadequacy of voluntary governance mechanisms may invite more prescriptive and burdensome regulatory interventions. A self-regulatory approach grounded in a layered framework enables developers to calibrate governance mechanisms to varying levels of risk, rather than imposing uniform and potentially overbroad restrictions. We will make this risk-based calibration more explicit in the revised paper, including the discussion currently in Section 4.1. In particular, we will clarify that the framework is modular by design: smaller labs and early-stage teams can satisfy core disclosure requirements by completing only the mandatory fields, while more comprehensive sections remain optional and are calibrated to higher-risk or higher-capability models.

---

> > ### Author Rebuttal · Reviewer_vJPe · 2026-04-04
> >
> > Thank you so much for your thoughtful response and for being willing to adjust your framework.
> >
> > I would like to thank the authors for their response.
> >
> > While I really appreciate the effort, I am still worried that these proposals might have adverse effect. For example, changing the data tracking requirement to checks makes it hard to know what the model is trained on and does not help us to do a proper research on top of it without understanding of what the model was really trained on.
> >
> > I still do not see how having such regulations helps us easier to collaborate.
> >
> > This will still make companies keep their models private.

---

### Official Review · Reviewer_kPeP · 2026-03-13

**Significance:** 4
**Argument Clarity:** 4
**Rating:** 5
**Confidence:** 3

**Questions:**

How do the authors envision repositories (e.g., Hugging Face) implementing the proposed Safety Card Template for OWFMs and the OWFM-tailored licensing in practice? In particular, how could such standards be adopted and enforced across the ecosystem, and what practical challenges might arise during deployment? What about statistical specification in learnware paradigm capturing model/agents capabilities beyond semantic description? (like Learnware of Language Models: Specialized Small Language Models Can Do Big)

**Alternative Views Section:**

Yes

**Compliance With Llm Reviewing Policy A Conservative:**

Affirmed.

**Discussion Potential:**

3

**Paper Summary:**

This position paper argues that current model cards are insufficient as governance mechanisms for open-weight foundation models (OWFMs). To support this claim, the paper analyzes 500 model cards hosted on Hugging Face, highlighting limitations in how safety-relevant information is documented and communicated to downstream users. The paper further argues that standard open-source licenses (OSLs) are poorly suited for OWFMs and may undermine the enforceability of acceptable use policies (AUPs).

Building on these observations, the authors propose a multi-layer governance framework that integrates three complementary artifacts: (i) model cards, (ii) acceptable use policies (AUPs), and (iii) licensing mechanisms. The paper also outlines directions for evolving these components into integrated safety artifacts, enabling a governance framework that coherently combines informational transparency, normative guidance, and legal enforceability.

**Position:**

Yes

**Position In Title:**

Yes

**Related Work:**

3

**Strengths And Weaknesses:**

### Strengths

1. The analysis of 500 model cards from Hugging Face provides strong empirical grounding for the paper’s argument. Besides, the authors provide detailed implementation proposals in the appendix, including a Safety Card Template for OWFMs and an OWFM-tailored licensing regime (e.g., the proposed Open-Weight Model Revision of Apache License 2.0).
2. The paper offers several insightful observations about structural tensions between existing governance mechanisms. For example, it highlights the incompatibility between permissive open-source licenses and the use-based restrictions introduced through acceptable use policies (AUPs), which is an important but often overlooked issue in current discussions of open model governance.
3. The paper proposes a structured governance framework that integrates model cards, acceptable use policies, and licensing mechanisms. This framing helps clarify the distinct roles of informational transparency, normative guidance, and legal enforceability in downstream governance. The inclusion of concrete implementation directions further strengthens the proposal.
4. The paper provides a relatively comprehensive discussion of alternative perspectives and potential concerns.

### Weaknesses

1. Limited discussion of practical implementation. While the proposed Safety Card Template for OWFMs and the OWFM-tailored licensing regime are conceptually appealing, the paper provides limited discussion of how repositories, developers, and downstream users would enforce these integrated governance artifacts at scale.

**Support:**

3

---

> ### Author Rebuttal · Authors · 2026-03-30
>
> We thank the reviewer for the insightful and constructive feedback. We especially appreciate the recognition of the empirical grounding and the structural clarity of our three-layer governance framework. Below, we provide a detailed response addressing each of the weaknesses [W#] and questions [Q#] raised.
>
> ---
>
> **[W1 / Q1]** We agree that practical implementation is a central challenge, and we clarify that our proposal is intended to be deployable through **incremental integration into existing repository infrastructures**, rather than requiring a fully centralized enforcement mechanism from the outset.
>
> Concretely, repositories such as Hugging Face can adopt the framework through three progressively implementable steps:
>
> (1) **Standardized metadata schemas**: The Safety Card can be integrated as a structured, machine-readable extension of existing model card metadata formats (e.g., JSON/YAML fields), enabling consistent disclosure of model provenance, risk characteristics, and usage constraints without changing existing developer workflows.
>
> (2) **License binding and interface-level enforcement**: The OWFM-tailored license (Appendix V, Section 10) can be operationalized through repository-level mechanisms such as gated access, click-through agreements, and API-based usage terms, which are mechanisms already partially in use for gated models on Hugging Face. Our proposal formalizes their alignment with AUPs and Safety Card disclosures.
>
> (3) **Compliance pull**: Rather than assuming perfect enforcement, the framework emphasizes **traceability and signaling,** enabling downstream developers and users to identify, assess, and mitigate risks, a necessity as major jurisdictions increasingly mandate such oversight. Safety Cards have already emerged as a vital tool for downstream information flow, facilitating compliance across the full supply chain. Our proposed template is explicitly aligned with such documentation obligations (Section 3.1.3), including MDF requirements for GPAI models under the EU AI Act and recent guidelines under Korea’s AI Framework Act, which require downstream deployers of open-weight models for high-impact use cases to verify model cards for compliance.
>
> In this sense, enforcement is not assumed to be absolute, but **distributed across the ecosystem**, combining repository-level controls with downstream incentives.
>
> ---
>
> **[Q2]** We thank the reviewer for raising the learnware paradigm (Tan et al., 2025), which we view as a valuable complement to our framework. The learnware paradigm associates each model with a **capability-level specification**, a functional fingerprint derived from model behavior that enables **privacy-preserving capability matching** between models and user tasks. Such specification-based representations can serve as a **machine-verifiable complement** to Safety Cards, grounding capability claims in data-driven signatures rather than self-reported benchmarks, thereby also reducing the risk of safetywashing identified in Section 4.2. Moreover, the learnware paradigm's emphasis on **decentralized expertise integration** (aggregating model contributions without exposing private training data) resonates with our framework's recognition that upstream influence disclosure is often constrained by data privacy, offering a complementary technical pathway for heritage tracking. We note that current learnware approaches focus on task-specific capability matching; extending them to capture safety-relevant properties (e.g., refusal behavior, adversarial robustness) is a promising direction we will flag in Section 5.2 and incorporate into the related work section.
>
> ---
>
> **Reference**
>
> Tan, Z.-H., et al. (2025). Learnware of Language Models: Specialized Small
> Language Models Can Do Big. *arXiv preprint arXiv:2505.13425*.

---

### Official Review · Reviewer_Q7px · 2026-03-23

**Significance:** 4
**Argument Clarity:** 4
**Rating:** 6
**Confidence:** 4

**Questions:**

- Similar to open-source software, the actual use and final application of OSFMs may not be traceable or effectively governed.  OSFMs may be fine-tuned or used as an ingredient to build substantially different models. It may be challenging to determine which model a given model was adapted from.

- OSFMs are probabilistic models by nature. How should this nature be factored into the licenses?

**Alternative Views Section:**

Yes

**Compliance With Llm Reviewing Policy A Conservative:**

Affirmed.

**Discussion Potential:**

3

**Paper Summary:**

The paper discusses an important issue of the mismatch among the practices surrounding the open-weight foundation models (OWFMs), their acceptable use policies (AUPs), and the open-source licenses (OSLs). The paper points out that the nature and usage of OWFMs are substantially different from previous open-source software. OSLs designed for open-source software are not compatible with the needs and requirements of OWFMs and even create legal conflicts with the model's AUPs. The paper also discusses the situation in the current model cards, where the AUPs are inconsistently disclosed and biased towards the model developers' interests. The paper calls for an overhaul of the model card and licensing of OWFMs to meet its novel and unique needs. The paper provides legal templates adapted for OWFMs

**Position:**

Yes

**Position In Title:**

Yes

**Related Work:**

4

**Strengths And Weaknesses:**

# +

- The paper is very nicely written. The logic is clear and the arguments are substantiated. The problem being discussed is relevant, timely, and important. The efforts and contributions toward this issue are valid and valuable.

- Writing is comprehensive.  Abundant examples and cases are provided, and evidence is added at appropriate places. The paper is solid.

---

# -

- The layout is a bit dense. With the clear flow of the logic, the elaboration can be made lighter and more concise, which I think may help attract more readers and retain their attention span for longer.

**Support:**

4

---

> ### Author Rebuttal · Authors · 2026-03-30
>
> We thank the reviewer for the thoughtful and positive assessment of our work. We particularly appreciate the recognition of the clarity, relevance, and importance of the problem. Below, we provide a detailed response addressing each of the weaknesses [W#] and questions [Q#] raised.
>
> ---
>
> **[Q1]** We agree that full traceability in open-weight foundation models is inherently challenging due to fine-tuning and compositional reuse. This important concern is addressed in our framework through a **dual-governance approach**. First, **Section 2 of the OWFM Safety Card Template (Appendix IV)** requires developers to disclose model heritage and upstream influences. Second, **Article 4(b) of the OWFM-tailored license (Appendix V)** mandates that creators of a “Derivative Model” include prominent notices in an accompanying Model Card specifying all modifications. Together, these provisions provide **structured provenance signals** and support a **“chain of custody”** across complex, multi-layered AI supply chains.
>
> ---
>
> **[Q2]** The probabilistic nature of OWFMs does not fundamentally undermine the scope or enforceability of a license because the license governs the model as a **functional asset**, not its individual outputs. Legally, an OWFM is a collection of weights, parameters, and software code. While the application of these weights results in non-deterministic outputs, the license attaches to the underlying technology, not the specific results it produces. Just as a license for a chemical catalyst remains enforceable regardless of the specific reaction yield, an OWFM license governs the reproduction, distribution, and modification of the model itself.
>
> In this sense, the probabilistic nature of OWFMs is a **functional characteristic of the system**, not a boundary of the legal grant. This further motivates our **layered approach**, where stochastic risks are addressed through complementary **informational** (model cards) and **normative** (AUPs) mechanisms alongside licensing.
>
> ---
>
> **[W1]** We appreciate this feedback. The current density reflects our attempt to comprehensively cover **three distinct governance artifacts** within a unified framework. We will improve clarity and conciseness by streamlining overlapping explanations and simplifying presentation where possible.

---

> > ### Author Rebuttal · Reviewer_Q7px · 2026-04-03
> >
> > I appreciate the authors' response.  These are fundamental questions that need more proactive thoughts than a "solution", which have already been factored in in my original review. I have read through the reviews and discussions, which are all positive. Good effort and good luck.

---

### Decision · Program_Chairs · 2026-04-30

**Decision:**

Accept (regular)

**Comment:**

The paper presents a strong and timely argument with compelling evidence from a study of 500 model cards on huggingface. The paper gives insightful observations and provides actionable advances and plans that can help fill the gap in existing artifacts used for downstream model governance.